# Swaying slower reduces the destabilizing effects of a compliant surface on voluntary sway dynamics

Dimitrios A. Patikas[1]*, Anastasia Papavasileiou[1], Antonis Ekizos[2,3], Vassilia Hatzitaki[4], Adamantios Arampatzis[2,3]

**1** School of Physical Education and Sport Science at Serres, Aristotle University of Thessaloniki, Thessaloniki, Greece, **2** Department of Training and Movement Sciences, Humboldt-Universität zu Berlin, Berlin, Germany, **3** Berlin School of Movement Sciences, Humboldt-Universität zu Berlin, Berlin, Germany, **4** School of Physical Education and Sport Science, Aristotle University of Thessaloniki, Thessaloniki, Greece

* dpatikas@auth.gr

**Data Availability Statement:** All relevant data are within the paper and its Supporting Information files.

## Abstract

The ability to control weight shifting (voluntary sway) is a crucial factor for stability during standing. Postural tracking of an oscillating visual target when standing on a compliant surface (e.g. foam) is a challenging weight shifting task that may alter the stability of the system and the muscle activation patterns needed to compensate for the perturbed state. The purpose of this study was to examine the effects of surface stability and sway frequency on the muscle activation of the lower limb, during visually guided voluntary postural sway. Seventeen volunteers performed a 2-min voluntary sway task in the anterior-posterior direction following with their projected center of pressure ($CoP_{AP}$) a periodically oscillating visual target on a screen. The target oscillated at a frequency of 0.25 Hz or 0.125 Hz, while the participants swayed on solid ground (stable surface) or on a foam pad (unstable surface), resulting in four experimental conditions. The electromyogram (EMG) of 13 lower limb muscles was measured and the target–$CoP_{AP}$ coupling was evaluated with coherence analysis, whereas the difference in the stability of the system between the conditions was estimated by the maximum Lyapunov exponent (MLE). The results showed that slower oscillations outperformed the faster in terms of coherence and revealed greater stability. On the other hand, unstable ground resulted in an undershooting of the $CoP_{AP}$ to the target and greater MLE. Regarding the EMG data, a decreased triceps surae muscle activation at the low sway frequency compared to the higher was observed, whereas swaying on foam induced higher activation on the tibialis anterior as well. It is concluded that swaying voluntarily on an unstable surface results in reduced $CoP_{AP}$ and joint kinematics stability, that is accomplished by increasing the activation of the distal leg muscles, in order to compensate for this perturbation. The reduction of the sway frequency limits the effect of the unstable surface, on the head and upper body, improves the temporal component of coherence between CoP and target, whereas EMG activity is decreased. These findings might have implications in rehabilitation programs.

**Funding:** This work was supported by the German Academic Exchange Service (DAAD) "GGP-Age", 57339989 (https://www.daad.de/en/). The funder played no role in the study design, data collection and analysis, decision to publish, or preparation of the manuscript.

**Competing interests:** The authors have declared that no competing interests exist.

## Introduction

Although bipedal stance is an essential and seemingly simple task, maintaining balance under various circumstances, as for example in the presence of external mechanical perturbations, is a rather complex issue [1,2]. The challenge for the central nervous system (CNS) is to integrate sensory input mainly from visual, proprioceptive and vestibular sources, and to create appropriate motor commands that take into account all environmental constraints and requirements. Challenging the sensorimotor system, including perturbations during different balance tasks, has been proposed as an efficient approach to understand neuromuscular control mechanisms for maintaining stability [3–5]. Understanding the adaptive responses of the motor system to cope with challenging balance conditions can improve our knowledge in order to develop successful exercise interventions to improve stability and to decrease risk of falls.

One of the most common strategies to introduce external perturbations is the use unstable surfaces with high viscoelasticity [6,7]. The viscoelastic properties of such surfaces reduce the effectiveness of transferring the ankle torque to the ground in order to adjust the body's position [8–10], and have been frequently used as means to improve postural stability in older individuals [7,11] and patients [12,13]. Previous research supports the notion that standing on a compliant surface decreases the reliability of sensory input from the plantar mechanoreceptors [3,14,15] and changes the contribution of visual, vestibular and somatosensory information to control balance [16]. However, there is still much to be discovered on how the system reacts and adapts to this type of perturbation in terms of muscle activation. This knowledge, for example, has implications in the capacity of the CNS to control the center of pressure (CoP) accurately during voluntary tasks, such as forward/backward whole-body sway on stable or unstable surfaces.

Shifting body weight is necessary for everyday activities, such as gait initiation, getting up from a chair, and might be crucial under externally imposed sensory constraints (e.g. when avoiding an obstacle or stepping on a slippery surface). According to an observational study [17], incorrect weight shifting is the most prevalent cause of falling in older adults while reduced amplitude of voluntary sway is related to an increased risk of falling [18]. For this reason, the ability to perform fine and accurate adjustments of the CoP (voluntary body sway) has been used in the past as a rehabilitation tool for people with balance deficits [19–22]. Previous studies have shown that the frequency of voluntary sway in the anterior-posterior direction affects stability [23] and the spatiotemporal variability of the oscillations [24]. More specifically, stability, as evaluated by the margins of stability [25], was greater when the sway frequency increased [23], whereas variability was higher with increasing frequency [24]. However, it is still unexplored how the system responds to external perturbations–by changing for example the compliance of the support surface–during voluntary sway at low frequencies, close to or even lower than the natural (i.e. self-selected), voluntary sway frequency during standing [26,27]. Thus, it is possible that there is an interaction of oscillation frequency with the superimposed perturbations that influences differently the neuromuscular responses of the sensorimotor system.

Considering the above, the main purpose of this study is to manipulate the surface stability (i.e. standing on firm ground or on foam) and the execution speed (i.e. two different sway frequencies) examine the effects on manipulating surface stability and execution speed during voluntary, visually guided voluntary postural sway. It is expected that swaying at a lower frequency may improve coherence between the target and the CoP when standing on unstable ground and this might be accompanied with changes in the activation of the lower limb muscles. Therefore, it is hypothesized that a lower sway frequency may compensate for the instability of the system when swaying on unstable ground and this behavior might be reflected in the

EMG responses of the lower limb muscles and the ability to control more precisely the position of the CoP. The stability of the system will be quantified by the maximum Lyapunov exponent (MLE). The effects of the different sway conditions reflected on the CoP-target coupling, will be evaluated by means of coherence analysis. The aim of this study is to give useful information about how the system adapts to changes in the somatosensory input and to describe the compensatory strategies that the neuromuscular system develops when externally induced perturbations (standing on foam) are introduced. Furthermore, this study will describe the adaptation mechanisms employed when the time to process the sensory input is prolonged, i.e. during voluntary sway at a slower sway frequency. This information may be useful when applying voluntary sway in rehabilitation.

## Materials and methods

### Participants

The experiment was performed with the approval of the institution's (Humboldt University) ethics committee (approval code: HU-KSBF-EK_2018_0013) in accordance with the Declaration of Helsinki. The consent was informed in written form, and the study did not include any minors. Seventeen healthy adults (10/7 males/females, mean±SD age 32.1±5.8 years, height 175±8 cm, body mass 69.8±12.9 kg) were recruited. All participants had normal or corrected to normal vision, wore no orthotic insoles and none of them had a history of neuromuscular impairments or balance-related dysfunctions. Prior to their inclusion participants were informed about the experimental protocol and gave their written informed consent.

### Procedures

Prior to electrode placement, body mass and height were measured. Furthermore, foot length was calculated as the average distance between the tip of the toe and the calcaneus of both feet. Bipolar surface electrodes (sensor area 15 mm$^2$, wet gel Ag/AgCl, N-00-S, Ambu A/S, Denmark) were placed over 13 superficial lower limb muscles of the right side. The muscles measured were the gluteus medius, gluteus maximus, tensor fasciae late, rectus femoris, vastus medialis, vastus lateralis (VL), semitendinosus, biceps femoris, tibialis anterior (TA), peroneus longus, medial gastrocnemius (MG), lateral gastrocnemius (LG) and soleus muscle (SO). The inter-electrode distance was set at 2 cm and the electrodes were positioned according to the recommendations of the SENIAM project [28]. The skin was carefully prepared (shaving, abrasion with sandpaper and cleaning with alcohol solution) to assure good electrode-skin contact. Manual tests (brief muscle contractions) with real-time visual inspection of the raw EMG were performed to verify the electrode placement and to ensure minimum amount of crosstalk.

During the assessment the participants stood on a force platform (60×90 cm, Kistler, Winterthur, Switzerland). The following experimental paradigm was used to create a visually guided weight-shifting task for voluntary controlling the CoP in the anterior-posterior direction. A monitor (47-inch diagonal) was placed at eye level, 1.5 m in front of them and displayed 2 dots (yellow and red) with black background (Fig 1A). The yellow dot (feedback) showed in real-time the position of CoP at the anterior-posterior direction (CoP$_{AP}$). The red dot (target) represented the position of CoP$_{AP}$ that the participants were instructed to follow with the yellow dot. The movement of both dots was restricted to the vertical direction, in the middle of the screen. Upward or downward movement of the dots signified shifting of the CoP to the anterior or posterior direction, respectively.

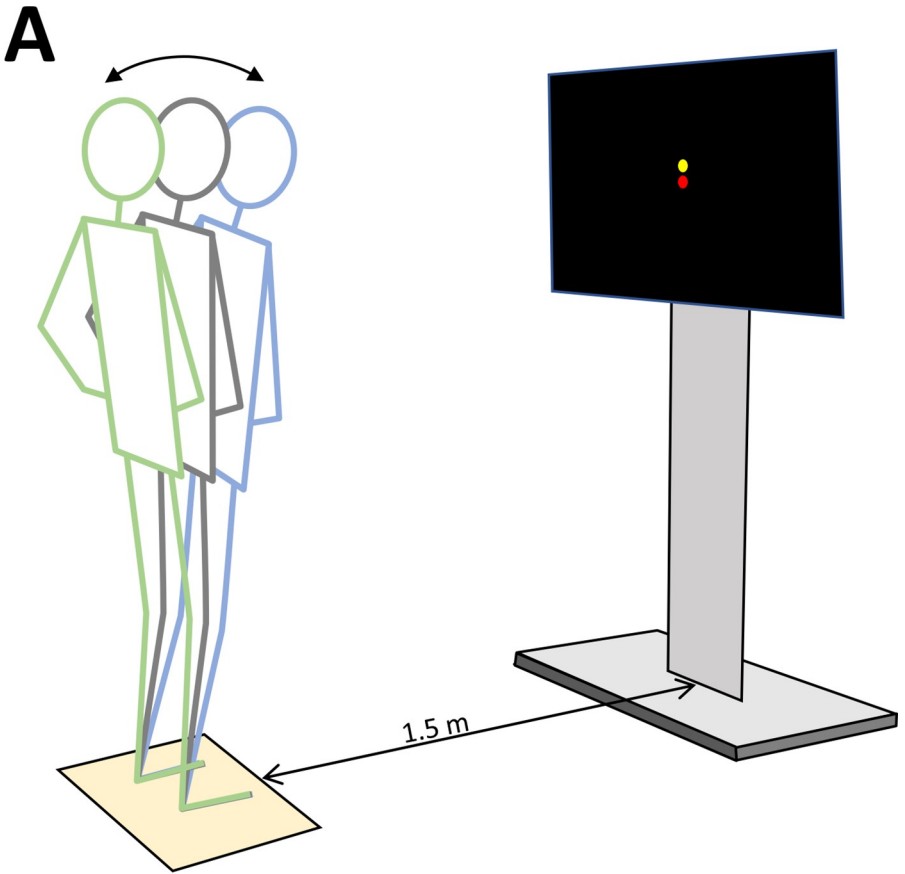

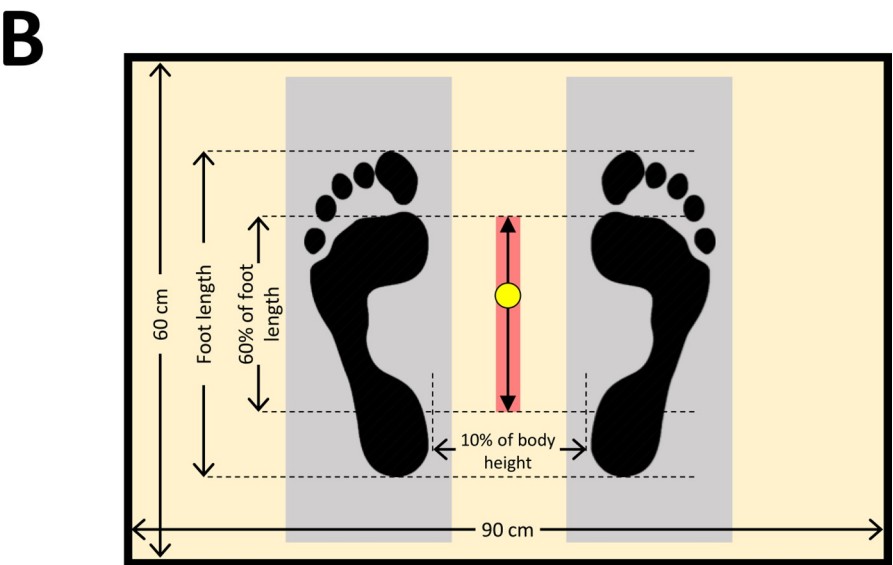

**Fig 1. Schematic illustration of the experimental setup.** A: Representation of a participant performing voluntary forward (blue) and backward (green) sway and watching on the monitor the target (red dot) that is moving up and down, and the feedback of his/her $CoP_{AP}$ (yellow dot), with the objective to match the two dots. B: Transverse view of foot placement on the force platform and the position of the CoP (yellow dot). Red shaded area indicates the oscillating range of the target (60% of foot length). Areas shaded in grey designates the position of the foam pads for the trials on unstable ground.

## Postural tasks

The participants stood on both feet with their hands at their waist (akimbo position). The distance between the two medial malleoli was set at 10% of the body height. The postural tasks consisted of visually guided body sway in the anterior-posterior direction. The target (red dot) was a time-series sinusoidal signal, generated by a sine wave with a fixed time period for each condition (either 4 or 2 s, to create an oscillation at 0.25 or 0.125 Hz, respectively), with a sample frequency of 50 Hz (i.e. one data point every 20 ms, which resulted in 8000 and 4000 samples per cycle for the 0.25 and 0.125 Hz conditions, respectively). The target (red dot) moved vertically in a sinusoidal pattern and the participant was instructed to match the red dot with the yellow one, by swaying his/her torso forward or backward, without flexing the hips or knees. The amplitude of the target movement (red dot) was set at 60% of the foot length, with zero representing the midpoint of the $CoP_{AP}$ range when the participant was leaning as much as possible anteriorly and posteriorly without moving the feet from the ground (Fig 1B).

Four postural tasks varying in sway frequency and ground stability were performed in random order. Two sway frequencies were selected; one natural [26,27] with a period of 4 s (0.25 Hz) and one lower, with a longer of 8 s (0.125 Hz). For each frequency, voluntary sway was recorded on rigid ground and on foam surface. A familiarization session before the measurement was performed and prior to the measurement, room lights were dimmed. Each trial lasted 2 minutes with 2–3 minutes interval in-between. The session, including the subject preparation (15 minutes), lasted no longer than one hour, considering the set-up for ultrasound recordings and two additional conditions (data not presented here).

## Data acquisition

The target signal was created and displayed on the monitor with custom made MATLAB scripts (version 2014b, Math Works Inc, USA), while an interface was created for triggering and synchronizing all devices with a single pulse. The force platform signal was digitized with a 14-bit resolution A/D card (NI USB-6009, National Instruments, USA) at 1000 Hz sampling rate and the anterior-posterior component was normalized to the foot length and was returned as input to the monitor for the vertical position of the yellow dot, with the full height of the screen representing 100% of foot length. The refresh rate of the dots was set at 50 fps.

The EMG signals were captured with a wireless EMG system (myon m320, myon AG, Schwarzenberg, Switzerland). The signal was pre-amplified (gain: 500, input impedance: 2 MΩ, bandwidth: 5–500 Hz) and transmitted at 12-bit resolution with 1000 Hz sampling frequency. All digitized signals were stored for further processing.

## Data processing

$CoP_{AP}$ signal was filtered with a $4^{th}$ order Butterworth low-pass filter with cutoff frequency at 25 Hz, and the first cycle was omitted from the analysis. The $CoP_{AP}$-target coupling was evaluated using the spectral coherence analysis which represents the amount of correlation between the two signals on the frequency domain from 0 to 1 Hz. Both target and $CoP_{AP}$ signals were interpolated at a sampling frequency of 64 Hz which responded to a frequency resolution of 0.0625 Hz when assessing the fast-Fourier transform. Three variables were analyzed at the specific frequencies that the task was executed (i.e. either at 0.125 Hz or 0.25 Hz for the slow or natural sway, respectively): the spectral coherence as a measure of the correlation between the two signals (target and $CoP_{AP}$) in the frequency domain, spectral phase as a temporal measure of the phase lag between the signals, and the spectral gain which reveals spatial information about the amplitude of the two signals (values over 1 designate $CoP_{AP}$ values above and below the target peaks and valleys, respectively). The spectral phase was expressed in % of the sway

cycle, with 0 being interpreted as the absolute synchronization between the signals and negative values as a delayed $CoP_{AP}$ relative to the target signal.

Local dynamic stability represents the ability of a system to maintain its movement pattern despite intrinsic and extrinsic perturbations [29–31]. The local dynamic stability of the system in the current study was assessed through the maximum finite-time Lyapunov exponent (MLE), which quantifies the rate of divergence of nearby trajectories in the reconstructed state space [32,33]. Our analysis followed the procedure as described in a previous study [34].

In the present study we calculated the MLE in the CoP and coordinate data of the markers placed in different parts of the body. For the CoP, data acquisition was performed at 2,000 Hz and MLE has been calculated on the norm of the anterio-posterior and mediolateral axes. The original time-series have been filtered using a 4[th] order Butterworth low-pass filter with a cut-off frequency of 20 Hz and were consequently down-sampled to 20,000 data points. The 3-dimensional coordinate data of the ankles (lateral malleoli), knees (lateral epicondyles), hips (greater trochanters), spine (7[th] cervical vertebra) and head (4 markers around the head placed on a headband) were acquired at 250 Hz and have been filtered in the same manner. All kinematic data were subsequently down-sampled to 15,000, and we calculated the midpoint between the two sides of the body and the midpoint of the 4 head markers. The 3-dimensional coordinates of these virtual markers in addition to the marker at spine were used for the MLE calculation. The norm of all axes has been used. Due to the standardized constant movement of the target signal and overall trial time (i.e. 120 seconds) the number of cycles (i.e. 30 cycles for the natural frequency and 15 for the slow frequency) was the same for all participants and no interpolation of the time-series was needed. To reconstruct the state space from the one dimensional time series, we used delay-coordinate embedding [35] as follows:

$$S(t) = [z(t), z(t + \tau), \ldots, z(t + (m - 1)\tau)], \tag{1}$$

with S(t) being the $m$-dimensional reconstructed state vector, z(t) the input 1D coordinate series, τ the time delay and $m$ the embedding dimension. Time delays were selected based on the first minimum of the Average Mutual Information function [36]. For these data $m = 3$ was sufficient to perform the reconstruction, similarly to previous studies examining human movement [29,34]. Individually selected time delays were chosen by averaging the outcome delays deriving from both trials performed by the participants [29,37]. Values of τ were slightly lower in the slow condition ($CoP_{AP}$: 0.11–0.19, coordinates: 0.14–0.33 of one cycle) compared to the natural condition ($CoP_{AP}$: 0.20–0.25, coordinates: 0.17–0.35 of one cycle).

Further, the average divergence of each point's trajectory to its closest neighbor was calculated, using the Rosenstein algorithm [38]. The resulting MLE was calculated based on the delay of each participant. That ensured the standardization of the calculation for the MLE across individuals, due to the first pick in the resulting divergence curves corresponding to 0.5 delay. As such, the final MLE value was calculated as the slope of the average divergence curves' linear fit corresponding to 0.5 of the individuals' delay value (i.e. the most linear part of the curve).

The EMG signals were filtered (Butterworth 4[th] order bandpass filter from 10 to 450 Hz), fully rectified and smoothed using a low-pass filter (Butterworth 4[th] order low pass filter at 5 Hz). To remove the baseline activation of each muscle the minimum EMG was subtracted from the filtered signal and all values were normalized to the mean of the trial. The start and end of each sway cycle was identified when the $CoP_{AP}$ moved from negative (posterior) to positive (anterior) values and crossed the zero line (Fig 2). Each cycle was interpolated to 200 data points and the average of all cycles was calculated, after excluding the first sway cycle.

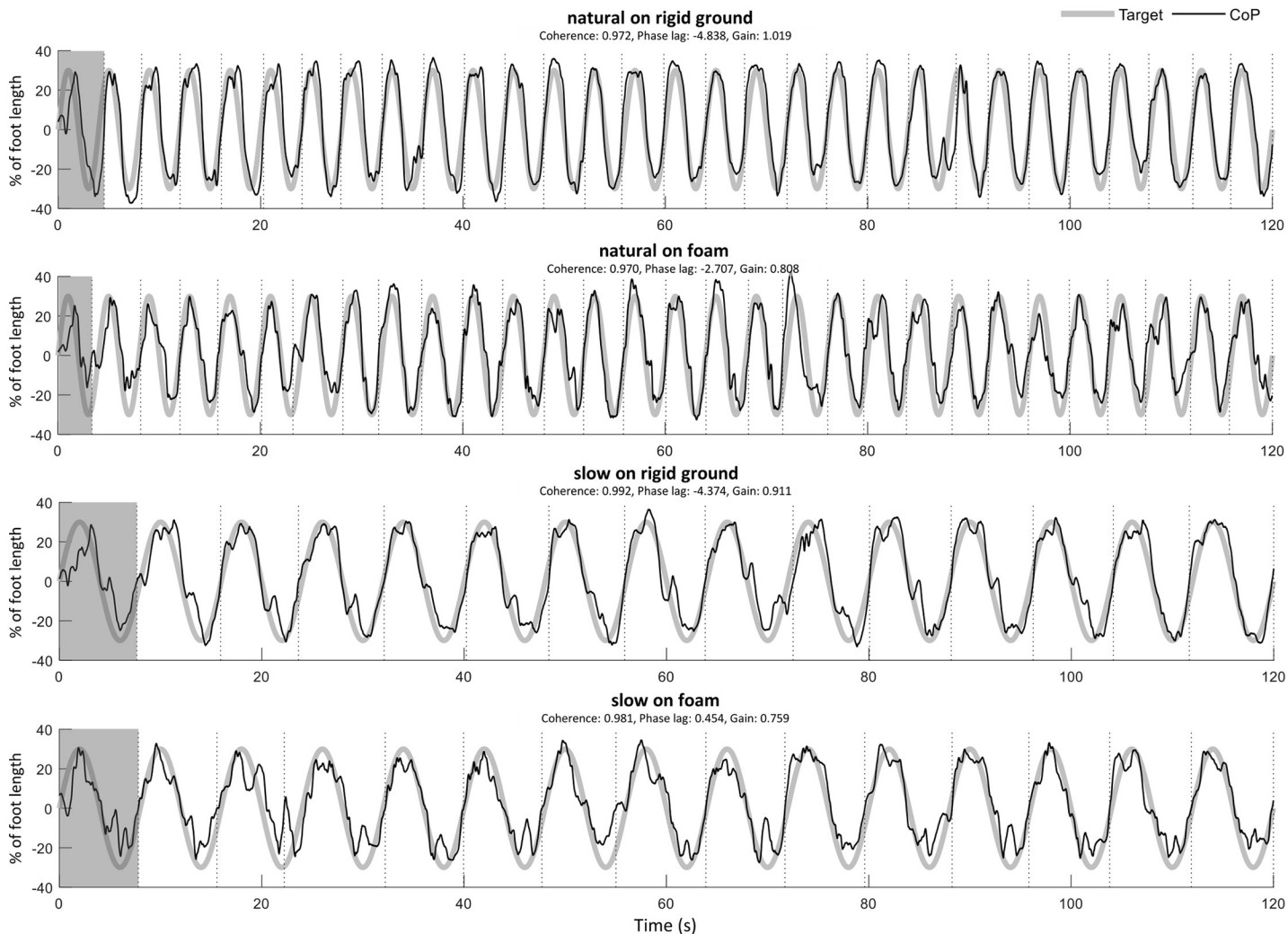

**Fig 2. Data from typical subject.** Target (thick gray line) and $CoP_{AP}$ (thin black line) are shown during the four conditions of the voluntary postural sway on rigid ground or on foam and with slow or natural sway frequency. Target-$CoP_{AP}$ coherence, phase lag (% of sway cycle) and gain for each condition are displayed on the middle top of the graphs, and values on vertical axis are presented as percentage of foot length. The vertical dotted lines represent the start and end of each sway cycle. Grey shaded area designate the first cycle that has been excluded from the analysis.

## Statistical analysis

Data are reported as mean±SD. For the spectral coherence analysis (coherence, phase lag and gain) and MLE, a two-way ANOVA design for repeated measures was assessed to detect the effects of ground surface and sway frequency. The Scheffé post-hoc test was performed when the level of significance was reached. To compare the effects of the different conditions on the $CoP_{AP}$ and EMG curves, paired t-test comparisons were assessed for all combinations using one-dimensional statistical parametric mapping [39]. For this purpose, the open-access SPM1D code for MATLAB was used (www.spm1d.org, v. 0.4). The level of significance α was set at 0.05 for all analyses.

## Results

A typical example of the $CoP_{AP}$ with the target signal in all four conditions is shown in Fig 2. The MLE (Table 1) of all variables showed a significant main effect for both factors (p<0.01

**Table 1. MLE values and statistical analysis for COP and kinematics.**

| | | Rigid ground | | | Main effect for surface Main effect for frequency Interaction surface×frequency | |
|---|---|---|---|---|---|---|
| | | | Foam | | F-values | p-values |
| CoP | Natural (0.25 Hz) | 13.7±1.7 | 14.4±1.4 | | $F_{1,16} = 11.8$ $F_{1,16} = 56.2$ $F_{1,16} = 0.03$ | p = 0.003 p<0.001 p = 0.860 |
| | Slow (0.125 Hz) | 10.0±1.4 | 10.8±1.9 | | | |
| Ankle | Natural (0.25 Hz) | 9.4±1.2 | 10.0±1.4 | | $F_{1,16} = 24.7$ $F_{1,16} = 45.6$ $F_{1,16} = 1.8$ | p<0.001 p<0.001 p = 0.199 |
| | Slow (0.125 Hz) | 6.7±1.3 | 7.7±1.4 | | | |
| Knee | Natural (0.25 Hz) | 9.3±1.1 | 11.2±1.1 | *** | $F_{1,16} = 52.4$ $F_{1,16} = 200.1$ $F_{1,16} = 8.8$ | p<0.001 p<0.001 p = 0.009 |
| | Slow (0.125 Hz) | 6.0±0.9 | 6.8±1.1 | n.s. | | |
| Hip | Natural (0.25 Hz) | 9.6±1.0 | 10.7±1.1 | ** | $F_{1,16} = 13.9$ $F_{1,16} = 243.4$ $F_{1,16} = 4.7$ | p = 0.002 p<0.001 p = 0.046 |
| | Slow (0.125 Hz) | 6.4±0.9 | 6.9±1.1 | n.s. | | |
| Spine | Natural (0.25 Hz) | 9.8±1.6 | 11.1±1.4 | *** | $F_{1,16} = 16.2$ $F_{1,16} = 105.7$ $F_{1,16} = 12.8$ | p<0.001 p<0.001 p = 0.002 |
| | Slow (0.125 Hz) | 6.3±1.0 | 6.7±1.0 | n.s. | | |
| Head | Natural (0.25 Hz) | 9.3±2.3 | 10.9±2.3 | ** | $F_{1,16} = 14.6$ $F_{1,16} = 57.0$ $F_{1,16} = 6.6$ | p = 0.002 p<0.001 p = 0.021 |
| | Slow (0.125 Hz) | 6.2±0.1 | 6.5±1.6 | n.s. | | |

Mean±standard deviation of the MLE for COP and norm coordinates at ankle, knee, hip, spine and head. Results of the 2-way ANOVA for each variable (F- and p-values for the main effects and interaction) are shown in the last two columns. Asterisks demonstrate significant difference for the post-hoc Scheffé test (n.s.: non-significant difference, p>0.05

**: p<0.01

***: p<0.001) between the rigid ground and foam for each frequency, when the interaction reached the level of significance.

for surface and frequency). The surface by frequency interaction was not significant for the CoP and the ankle joint (p>0.05). The significant (p<0.05) surface by frequency interaction in the rest of the variables and the respective post-hoc test, revealed increased MLE when standing on foam only for the natural sway frequency (p<0.01), whereas no statistically significant change was observed at the slower sway frequency (p>0.05).

Coherence of the target and $CoP_{AP}$ signals (Fig 3A) was significantly higher in the slow sway compared to the natural (p = 0.001). The unstable surface did not affect coherence (p = 0.109) and the interaction between sway frequency and surface was not significant (p = 0.858). Regarding the phase lag between the two signals (Fig 3B) the slower sway resulted in values closer to zero, whereas the natural sway frequency revealed significantly lower values, which shows a delayed response of the $CoP_{AP}$ relative to the target motion (p<0.001). Similar to the coherence, the phase lag did not show any significant differences (p = 0.126) between stable and unstable condition and the interaction between the two factors was also not significant (p = 0.265). The gain (Fig 3C) was significantly lower (under-shooting of $CoP_{AP}$ relative to the target) in the slow compared to the natural sway (p<0.001) and when on the foam compared to the rigid ground (p<0.001). However, the interaction between the factors frequency and surface was not statistically significant (p = 0.065).

As shown in Fig 4, when comparing the slow vs. the natural sway condition on rigid ground or the rigid ground vs. foam during natural sway, the difference between the target signals did not differ significantly (p>0.05). On the contrary, during the 0–13, 30–62, and 81.5–100% of

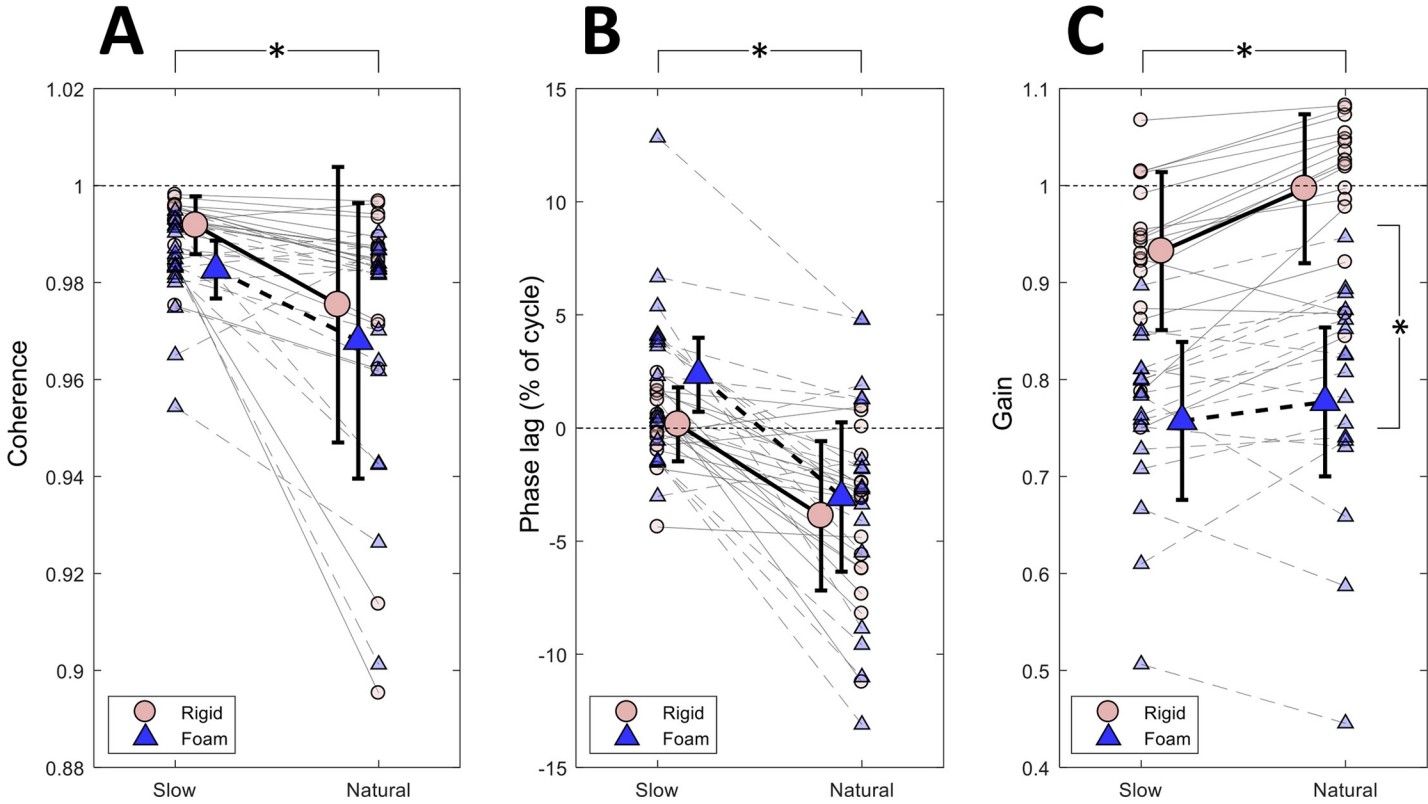

**Fig 3. Results of the coherence analysis.** Group results of the CoP$_{AP}$-target coherence (A), phase lag (B) and gain (C) for the slow and natural voluntary sway on stable (rigid) and unstable (foam) ground. Pink circles connected with continuous lines and blue triangles connected with dashed lines represent the condition of sway on rigid ground and foam, respectively. Small and large symbols (circles and triangles) represent data of each individual and group means, respectively. Vertical lines designate one standard deviation and asterisks express the presence of significant main effect (p<0.05).

the sway cycle, the target signals were significantly different between slow and natural sway on foam, due to a shift to the right for the natural relative to the slow sway condition. Similarly, during the 0–17.5 and 44.5–66.55 of the sway cycle, the target signals were significantly different between slow sway on rigid ground and foam, due to a shift to the left for the foam relative to the rigid ground surface.

Regarding the EMG recordings, a visual representation of the EMG amplitude for the 13 examined muscles of a typical subject during the voluntary sway in all conditions, is shown in Fig 5. On average, the results of EMG showed that gluteus maximus and medius, as well as tensor fascia latae had no observable phasic activation during any of the tasks (Fig 6). The rest of the muscles demonstrated phasic behavior, with activation when the CoP$_{AP}$ was at the extreme anterior or posterior phase of the sway cycle (i.e. transition phase from on direction to the other). Most remarkable differences between the conditions were observed at the distal muscles. According to the SPM analysis, slow compared to natural sway on rigid ground showed lower EMG activity for the MG and SO during 96–98.5% and 93–97.5% of the sway cycle, respectively. These differences were present for longer portion of the sway cycle (MG: 0–1, and 95.5–99.5%; SO: 0–1, 3.5–4.5, 9–12, and 94.5–100%) and for more muscles when this comparison was made for the foam condition (LG: 0–3.5, 6–8, 8.5–13, and 94.5–100%; TA: 58–63.5, 64–68.5, and 71–76.5%; VL: 18–27.5, and 50.5–53.5%). Specifically, during slow voluntary sway, the LG (at 69–70, 75.5–82, 83–84, and 85.5–87%) and VL (at 89–91%) revealed significantly higher activation when standing on foam compared to standing on the rigid surface.

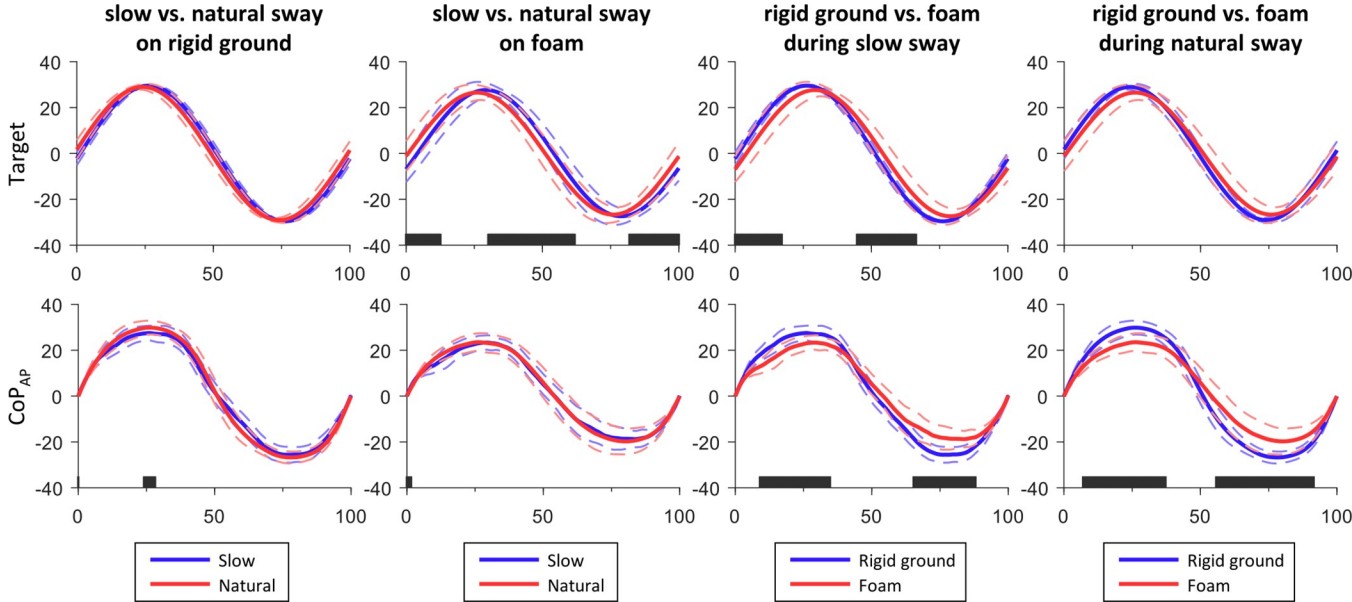

**Fig 4. Mean values of all participants for the target signal and CoP_AP during the voluntary sway.** Each column of graphs depicts one comparison between conditions. Dashed lines represent one standard deviation and black stripes at the horizontal axes illustrate the time of the sway cycle that there was a significant difference ($p < 0.05$) between the two conditions. Horizontal axis is normalized to the duration of each sway cycle of $CoP_{AP}$ and values for the vertical axis are expressed as percent of the foot length.

For the natural sway frequency, the higher values on foam compared to the rigid surface were detected on the TA (at 71–87%), LG (at 0–4, 71.5–76.5, 78.5–88, and 95.5–100%) and SO (at 0–2.5, 8–24, 25–29.5, 60–63, 66–71, 72–86, and 98–100%) muscles.

## Discussion

Voluntary sway on foam increases the instability of the system compared to rigid ground, making the system more unstable in both investigated frequencies. The increased instability resulted in reduced CoP-target gain (lower $CoP_{AP}$ values than the target) and in higher muscle activation of the distal muscles. On the other hand, decreasing the frequency of the voluntary sway resulted in lower muscle activation, better coupling of the $CoP_{AP}$ to the target, with stability differences (i.e. increased MLE) limited at the ankle and CoP level.

According to the current findings regarding the muscle activation patterns, it could be argued that during the voluntary visually guided sway the proximal muscles (i.e. muscles associated with the hip) demonstrate minimal or no phasic activity. This is supported by previous research arguing that the ankle strategy (implying no significant movement around the hip joint) can be retained at sway frequencies lower than 0.5 Hz [40]. Although it could not be excluded that the proximal muscles might be active even without movement on the hip to stabilize the trunk, our data give no evidence for such activation. On the other hand, the thigh and shank muscles were active during the first or second half of the cycle. Interestingly, ST, MG, LG and SO became active as the $CoP_{AP}$ moved forward (shortly before $CoP_{AP}$ crossed the midpoint), with the acting forces serving to decelerate the body's inertia and to initiate the backward sway (moving to a more plantar flexed position), as soon as the maximum anterior position (dorsiflexion) is reached. Likewise, the RF, VM, VL and TA act in the same manner, but in the opposite direction during the second half of the sway cycle. Based on the EMG data of the current study, it is evident that the CNS activates the muscles primarily as dampening

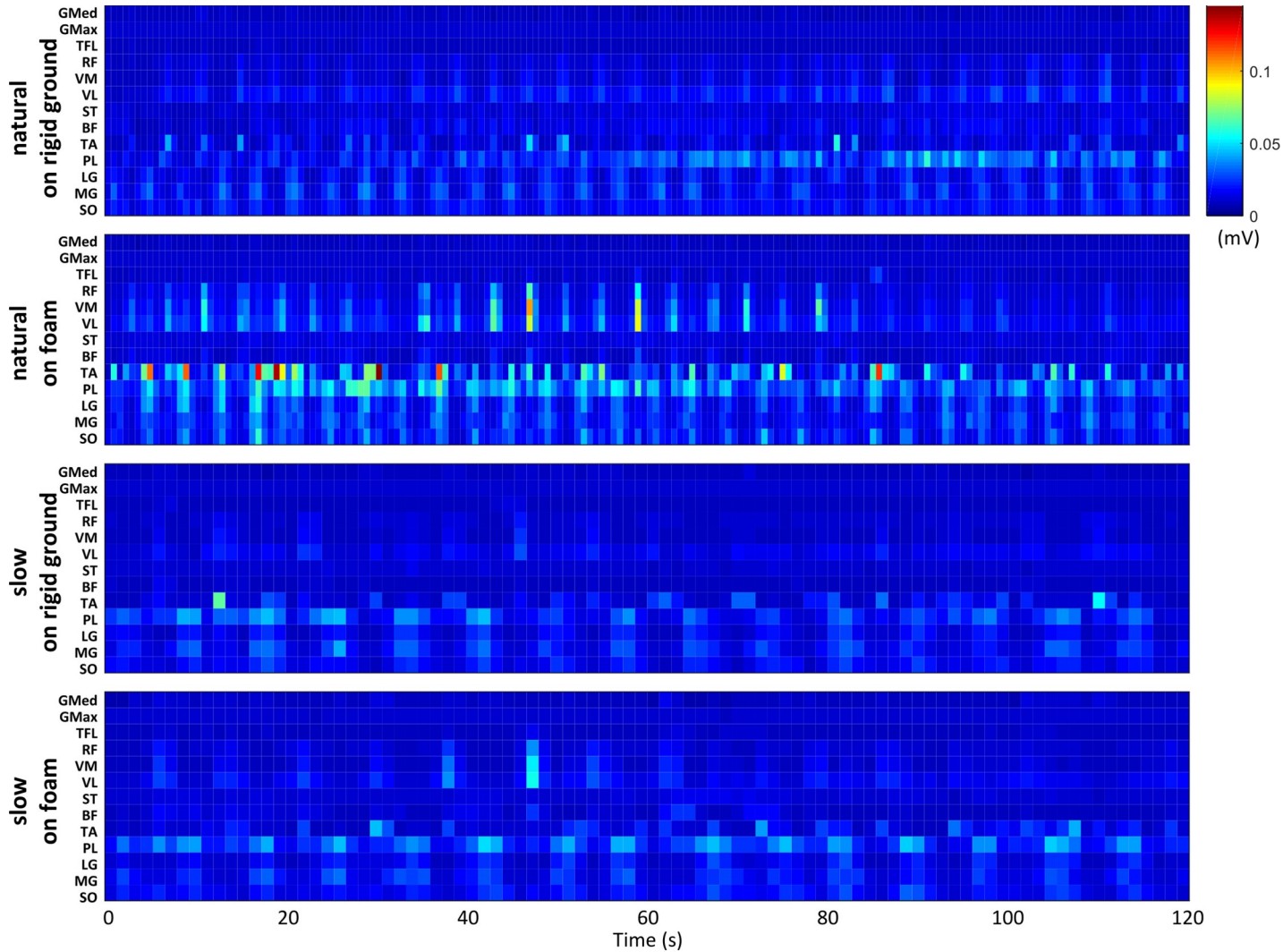

**Fig 5. Typical example of EMG signals at all four conditions of the 120-s voluntary sway.** Each of the four graphs represents one condition as described on the left side of the graph. The color of each cell represents the mean EMG amplitude for every 1/6 of the sway cycle. All muscles are shown at each row with the following abbreviations: gluteus medius (GMed), gluteus maximus (GMax), tensor fasciae late (TFL), rectus femoris (RF), vastus medialis (VM), vastus lateralis (VL), semitendinosus (ST), biceps femoris (BF), tibialis anterior (TA), peroneus longus (PL), medial gastrocnemius (MG), lateral gastrocnemius (LG) and soleus muscle (SO)].

elements to control the movement and secondary as active components to produce force in the direction of movement. This dampening function is in agreement with previous findings during slow-frequency voluntary sway and has been attributed to the limited capacity of the passive stiffness components to stabilize the body [41]. Furthermore, it could be argued that the system seems to function with two basic muscle groups (i.e. muscles that act synergistically comprising a muscle synergy), and two motor primitives (i.e. activation patterns). Similar muscle synergies that act reciprocally have been previously reported, even during faster (1 Hz) voluntary body sway [42] or during voluntary sway towards one direction (forward or backward) [42,43]. This distinction of two muscle synergies, reduces the dimensionality and thus complexity, making the system easier to control, organize and manipulate [44]. However, altering the environmental constraints, may increase the task complexity that could introduce new muscle synergies to the system [45,46].

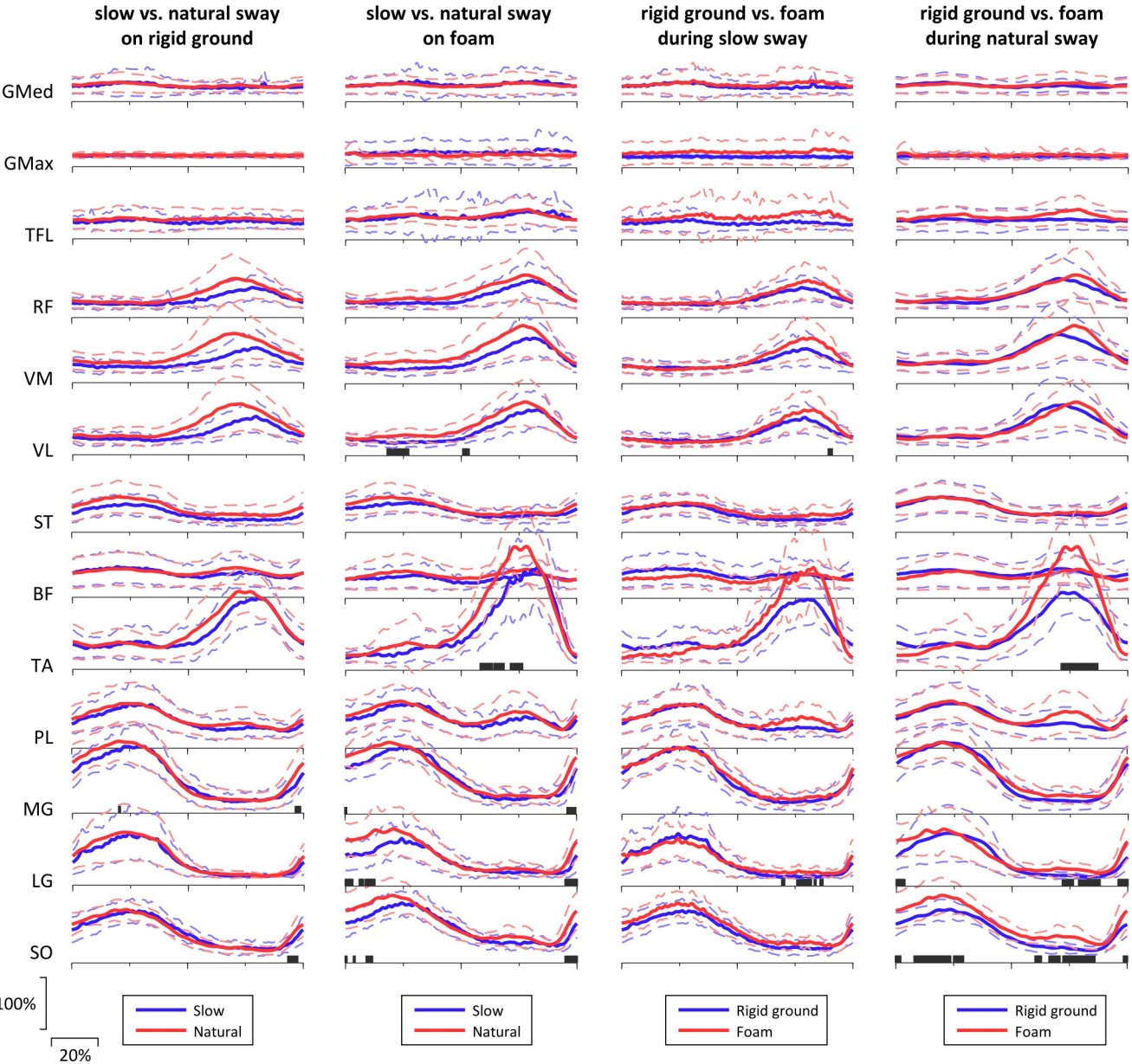

**Fig 6. Pairwise comparisons between the four voluntary sway conditions for the mean EMG amplitude.** See legend of Fig 5 for abbreviations of muscle names. Dashed lines represent one standard deviation and black stripes at the horizontal axes illustrate the time of the sway cycle that there was a significant difference ($p<0.05$) between the two conditions. Values for EMG are expressed in % of the mean EMG during the sway cycle and for time as % of the sway cycle.

The experimental paradigm used in the present study was selected to challenge the stability of the human system and to introduce external perturbations during the voluntary sway task, by changing the ground compliance. Indeed, MLE increased when standing on foam revealing higher instability, with less stable trajectories in time for the CoP and the ankle, knee, hip, spine and head kinematics. This could be attributed to the premise that standing on a compliant surface impedes the direct force transfer to the ground and reduces the quality of sensorimotor information to the CNS [47,48]. However, the stability of the upper body (head and spine) as well as that of the hip and knee joints was not affected by the unstable surface at the

low sway frequency. Earlier studies have supported the notion that the movement of the head and trunk during perturbations is controlled by vestibular, visual and proprioceptive inputs, independent of the ascending information from peripheral body segments [49]. This shows that the quasi-unpredictable situation on the ground (standing on foam), that tends to destabilize the body, has a reduced effect in the head and trunk in terms of stability, when the movement is executed slowly. This might have implications in rehabilitation programs or experimental setups that require more stable head movement during a task, in order to reduce the dependency on the visual and vestibular system.

The consequences of the increased instability of sway on the target–CoP coupling was the reduced gain shown on foam surface compared to the rigid ground condition, revealing that the participants were not capable to reach the target limits (peaks and valleys). The reduction in gain could be interpreted as a reduction in the limits of stability when standing on foam and could be attributed to the limited capacity to transfer forces to the ground [8–10] and the lower quality of somatosensory input [3,14,15]. Furthermore, there are indications of increased co-activation of the antagonist muscles when standing on foam and the CoP is at a posterior position (plantar flexion). This change is in agreement with previous studies that observed increased level of co-activation with increasing difficulty of postural tasks [50], which may act as compensatory mechanism, by increasing joint stiffness and thereby stability [51]. On the other hand, it counteracts towards the direction of the movement and thereby limits its range of motion.

An alternative explanation of the stronger temporal coupling during slow compared to natural sway and the stronger spatial coupling when standing on rigid ground compared to the foam surface could stem from the perceptual-motor reality of postural time-to-contact [52,53]. According to this concept, the temporal proximity to the margin of stability is reduced intuitively when a) sway velocity increases b) the base of support decreases and c) the degrees of freedom are reduced [54]. Regarding the findings of the present study, it could be argued that one reason for the weaker temporal coupling (greater phase lag) in the higher than the lower sway frequency might be the greater movement velocity that induced a reduction in the margins of stability. This is also reflected by the fact that MLE was increased only distally (CoP and ankle) when the surface of support became unstable at slower sway frequencies, whereas the consequences of the foam surface on the stability of the system emerged up to the head when swaying at higher frequencies.

The improved synchronization observed in the present study (lower phase lag between $CoP_{AP}$ and target) during slow oscillations, was characterized by reduced activation of the calf muscles. Although previous studies have shown that increasing the frequency of sway may involve the activation of more proximal muscles by moving the hip more actively [40], there is no evidence for such recruitment at least for the sway frequencies that have been tested. The improved temporal coupling is in agreement with studies performing voluntary periodic sway as fast as possible at a certain target range [55,56] and is in accordance with Fitts' law, which addresses a trade-off between accuracy and movement speed, i.e. slower movements are more accurate [57]. Another explanation for the improved temporal coupling during the slow sway is the greater stability observed on the head, which may introduce less bias from visual and vestibular sources. Furthermore, it has been suggested that when movement is slow, proprioceptive input might have a more prominent role to control movement and achieve stability [58], in contrast to faster movements, when the intrinsic mechanical properties of the system are of greater importance than the peripheral feedback [59].

In general, although the voluntary sway task was more challenging on the foam, especially during the higher frequency, no systematic EMG increase was shown on the proximal muscles. On the other hand, changes in EMG have been observed during the second half of the cycle,

when the RF, VM, VL and TA decelerate the body from moving backwards and initiate the forward motion. Therefore, it seems that there is a modular organization during voluntary sway that is retained when the system becomes more unstable in the perturbed condition. Furthermore, previous studies reported that distal muscles are more responsive to perturbations than proximal ones [58], possibly due to morphological and anatomical differences (i.e. large pennation angle, short fascicle and longer tendons) that reinforce sensitivity at low force levels [60]. The observed increased motor drive is in agreement with previous studies which support that unstable surfaces result in increased contraction speeds and higher motor output [61,62]. This supports the notion that muscle synergies are not simply defined as groups of muscles that act together, but as variables controlled by the CNS to co-vary depending on the task, in order to stabilize the body [43,63]. With the current experimental setup, we were not able to confirm any widening in the EMG activation time periods when changing the stability of the system. Recently it has been shown that when increasing unsteadiness during dynamic conditions (running or walking), muscle activation changes and creates a more "robust" motor output which results in developed strategies capable to cope with errors when required [59]. This discrepancy could be attributed to the fact that the task of voluntary sway, is less dynamic than walking or running, since the base of support is fixed and the range of motion of the involved joints during the movement is smaller. Therefore, under this condition, when fewer joints are involved with reduced degrees of freedom, the system may cope for stability with different strategies than the ones used during more dynamic movements such as walking or running. However, we found no indication that manipulation of the movement properties (i.e. changes in stability and sway frequency) created new synergies. It rather seems that the system modified the activation onset and amplitude of the already active muscles.

## Conclusions

In conclusion, standing on foam results in a more unstable CoP trajectory and body movement, and this is reflected to a higher muscle activation especially in the distal muscles. Slower execution of the voluntary sway limits the effect of the unstable surface, on the head and upper body, reduces the phase-lag between CoP and target, and exhibits a reduced EMG activity. However, it remains to be examined whether this behavior changes during aging or whether is different in patient populations. These findings can have implications in rehabilitation programs, depending on the goal of intervention and on which parameters (stability, accuracy, muscle activation) is necessary to be changed.

## Acknowledgments

The authors would to acknowledge the contribution of Dr. Arno Schroll for the creation of the interface for data collection and the hardware setup, Haris Sotirakis for contributing to the data analysis, as well as Dr. Sebastian Bohm, Dr. Alessandro Santuz and Victor Hugo Muñoz for their assistance in the recruitment of the participants.

## Author Contributions

**Conceptualization:** Dimitrios A. Patikas, Antonis Ekizos, Vassilia Hatzitaki, Adamantios Arampatzis.

**Data curation:** Dimitrios A. Patikas, Anastasia Papavasileiou, Antonis Ekizos, Vassilia Hatzitaki.

**Formal analysis:** Dimitrios A. Patikas, Anastasia Papavasileiou, Antonis Ekizos, Vassilia Hatzitaki, Adamantios Arampatzis.

**Funding acquisition:** Dimitrios A. Patikas, Vassilia Hatzitaki, Adamantios Arampatzis.

**Investigation:** Dimitrios A. Patikas, Antonis Ekizos, Vassilia Hatzitaki, Adamantios Arampatzis.

**Methodology:** Dimitrios A. Patikas, Anastasia Papavasileiou, Antonis Ekizos, Vassilia Hatzitaki, Adamantios Arampatzis.

**Project administration:** Dimitrios A. Patikas, Adamantios Arampatzis.

**Resources:** Anastasia Papavasileiou, Adamantios Arampatzis.

**Software:** Dimitrios A. Patikas, Antonis Ekizos, Vassilia Hatzitaki.

**Supervision:** Adamantios Arampatzis.

**Validation:** Dimitrios A. Patikas, Anastasia Papavasileiou, Antonis Ekizos, Vassilia Hatzitaki, Adamantios Arampatzis.

**Visualization:** Dimitrios A. Patikas, Anastasia Papavasileiou.

**Writing – original draft:** Dimitrios A. Patikas.

**Writing – review & editing:** Dimitrios A. Patikas, Anastasia Papavasileiou, Antonis Ekizos, Vassilia Hatzitaki, Adamantios Arampatzis.

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
