## [Decision Letter · Decision Letter 0]

30 Aug 2019

PONE-D-19-17729

Swaying slower reduces the destabilizing effects of a compliant surface on voluntary sway dynamics

PLOS ONE

Dear Dr. Patikas,

Thank you for submitting your manuscript to PLOS ONE. After careful consideration, we feel that it has merit but does not fully meet PLOS ONE’s publication criteria as it currently stands. Therefore, we invite you to submit a revised version of the manuscript that addresses the points raised during the review process.

We would appreciate receiving your revised manuscript by Oct 14 2019 11:59PM. To enhance the reproducibility of your results, we recommend that if applicable you deposit your laboratory protocols in protocols.io, where a protocol can be assigned its own identifier (DOI) such that it can be cited independently in the future. For instructions see: http://journals.plos.org/plosone/s/submission-guidelines#loc-laboratory-protocols

We look forward to receiving your revised manuscript.

Kind regards,

Yih-Kuen Jan, PhD

Academic Editor

PLOS ONE

Journal Requirements:

2. Please provide additional details regarding participant consent also in the ethics statement (currently you only describe consent given in the methods section). In the ethics statement in the online submission information, please ensure that you have specified (1) whether consent was informed and (2) what type you obtained (for instance, written or verbal, and if verbal, how it was documented and witnessed). If your study included minors, state whether you obtained consent from parents or guardians. If the need for consent was waived by the ethics committee, please include this information.

Reviewers' comments:

Reviewer's Responses to Questions

**Comments to the Author**

1. Is the manuscript technically sound, and do the data support the conclusions?

Reviewer #1: Yes

Reviewer #2: Partly

2. Has the statistical analysis been performed appropriately and rigorously? 

Reviewer #1: Yes

Reviewer #2: No

3. Have the authors made all data underlying the findings in their manuscript fully available?

Reviewer #1: Yes

Reviewer #2: Yes

4. Is the manuscript presented in an intelligible fashion and written in standard English?

Reviewer #1: Yes

Reviewer #2: Yes

5. Review Comments to the Author

Reviewer #1: The study is interesting; however, some minor concerns might need to be clarified.

1.The manuscript mention that seventeen volunteers performed a 2-min voluntary sway task (page 2), and each trail lasted 2 minutes with 2-3 minutes interval in between…no longer than one hour (page 7). How many trails for each volunteer in the study? Is there any pre-trial for subjects to familiar the device?

2.What is the function of the 2 dots (yellow and red) mentioned in the page 6? By the way, what is the purpose of the experimental design showed in figure 1 (A)?

3.What is target signal and how to capture and estimate?

4.The values showed in table 1 by the joint kinematics (Ankle, knee, hip, spine and haed). How these values be measured and estimated? And what are the meanings for these joint and head?

5.In figure 2, what is the unit in vertical axis?

6.In figure 3, the pink circle and blue triangle mean the average values?

7.Figure 5 show the typical example of EMG signal by the figure legend (page 27), however, it needs more explanations to realize the four figures mean.

8.Can the authors quantify a clear threshold or standard to define a sway frequency range for rehabilitation programs in the conclusion by the results?

Reviewer #2: See attached PDF review. Please see attached review. Please see attached PDF, which is my review. Perhaps now I have reached 100 characters.

6. PLOS authors have the option to publish the peer review history of their article (what does this mean?). If published, this will include your full peer review and any attached files.

Reviewer #1: No

Reviewer #2: No

---

## [Decision Letter · Decision Letter 1]

30 Oct 2019

PONE-D-19-17729R1

Swaying slower reduces the destabilizing effects of a compliant surface on voluntary sway dynamics

PLOS ONE

Dear Dr. Patikas,

Thank you for submitting your manuscript to PLOS ONE. After careful consideration, we feel that it has merit but does not fully meet PLOS ONE’s publication criteria as it currently stands. Therefore, we invite you to submit a revised version of the manuscript that addresses the points raised during the review process.

Please address the reviewer 2's comments and use the track changes to highlight the changes in the revised manuscript.  

We would appreciate receiving your revised manuscript by Dec 14 2019 11:59PM. To enhance the reproducibility of your results, we recommend that if applicable you deposit your laboratory protocols in protocols.io, where a protocol can be assigned its own identifier (DOI) such that it can be cited independently in the future. For instructions see: http://journals.plos.org/plosone/s/submission-guidelines#loc-laboratory-protocols

We look forward to receiving your revised manuscript.

Kind regards,

Yih-Kuen Jan, PhD

Academic Editor

PLOS ONE

Reviewers' comments:

Reviewer's Responses to Questions

**Comments to the Author**

1. If the authors have adequately addressed your comments raised in a previous round of review and you feel that this manuscript is now acceptable for publication, you may indicate that here to bypass the “Comments to the Author” section, enter your conflict of interest statement in the “Confidential to Editor” section, and submit your "Accept" recommendation.

Reviewer #1: All comments have been addressed

Reviewer #2: (No Response)

2. Is the manuscript technically sound, and do the data support the conclusions?

Reviewer #1: Yes

Reviewer #2: Partly

3. Has the statistical analysis been performed appropriately and rigorously? 

Reviewer #1: Yes

Reviewer #2: Yes

4. Have the authors made all data underlying the findings in their manuscript fully available?

Reviewer #1: Yes

Reviewer #2: Yes

5. Is the manuscript presented in an intelligible fashion and written in standard English?

Reviewer #1: Yes

Reviewer #2: Yes

6. Review Comments to the Author

Reviewer #1: The manuscript has been revised and all comments have been addressed completely by the authors' responses.

Reviewer #2: The authors Response to Reviews is good. Unfortunately, not much of that actually appears in the revised manuscript. The paragraph beginning Line 82 offers a prediction about EMG, but makes no other predictions, and says nothing about why it is useful to combine data on EMG with the various types of kinematic data. The impression continues to be that of a fishing expedition.

7. PLOS authors have the option to publish the peer review history of their article (what does this mean?). If published, this will include your full peer review and any attached files.

Reviewer #1: No

Reviewer #2: No

---

## [Decision Letter · Decision Letter 2]

25 Nov 2019

Swaying slower reduces the destabilizing effects of a compliant surface on voluntary sway dynamics

PONE-D-19-17729R2

Dear Dr. Patikas,

We are pleased to inform you that your manuscript has been judged scientifically suitable for publication and will be formally accepted for publication once it complies with all outstanding technical requirements.

With kind regards,

Yih-Kuen Jan, PhD, University of Illinois at Urbana-Champaign

Academic Editor

PLOS ONE

Additional Editor Comments (optional):

Reviewers' comments:

Reviewer's Responses to Questions

**Comments to the Author**

1. If the authors have adequately addressed your comments raised in a previous round of review and you feel that this manuscript is now acceptable for publication, you may indicate that here to bypass the “Comments to the Author” section, enter your conflict of interest statement in the “Confidential to Editor” section, and submit your "Accept" recommendation.

Reviewer #1: All comments have been addressed

Reviewer #2: All comments have been addressed

2. Is the manuscript technically sound, and do the data support the conclusions?

Reviewer #1: Yes

Reviewer #2: Partly

3. Has the statistical analysis been performed appropriately and rigorously? 

Reviewer #1: Yes

Reviewer #2: Yes

4. Have the authors made all data underlying the findings in their manuscript fully available?

Reviewer #1: Yes

Reviewer #2: Yes

5. Is the manuscript presented in an intelligible fashion and written in standard English?

Reviewer #1: Yes

Reviewer #2: Yes

6. Review Comments to the Author

Reviewer #1: The manuscript has been revised and all comments have been addressed completely by the authors'

responses.

Reviewer #2: In revising, the authors have made the minimum possible changes. By this choice, they have minimized the contribution of their study to the literature. I suppose that is their choice.

7. PLOS authors have the option to publish the peer review history of their article (what does this mean?). If published, this will include your full peer review and any attached files.

Reviewer #1: No

Reviewer #2: No

---

## [Editor Report · Acceptance letter]

3 Dec 2019

PONE-D-19-17729R2 

Swaying slower reduces the destabilizing effects of a compliant surface on voluntary sway dynamics 

Dear Dr. Patikas:

I am pleased to inform you that your manuscript has been deemed suitable for publication in PLOS ONE. Congratulations! Your manuscript is now with our production department. 

With kind regards,

on behalf of

Dr. Yih-Kuen Jan 

Academic Editor

PLOS ONE